



# Cell tracking of convective rainfall: sensitivity of climate-change signal to tracking algorithm and cell definition (Cell-TAO v1.0)

Edmund P. Meredith[1], Uwe Ulbrich[1], and Henning W. Rust[1]

[1]Institut für Meteorologie, Freie Universität Berlin, Berlin, Germany

**Correspondence:** Edmund P. Meredith (edmund.meredith@met.fu-berlin.de)

**Abstract.** Lagrangian analysis of convective precipitation involves identifying convective cells ("objects") and tracking them through space and time. The Lagrangian approach helps to gain insight into the physical properties and impacts of convective cells and, in particular, how these may respond to climate change. Lagrangian analysis requires both a fixed definition of what constitutes a convective object and a reliable tracking algorithm. Whether the climate-change signals of various object properties are sensitive to the choice of tracking algorithm or to how a convective object is defined has received little attention.

Here we perform ensemble pseudo global warming experiments at convection-permitting resolution to test this question. Using two conceptually different tracking algorithms, Lagrangian analysis is systematically repeated with different thresholds for defining a convective object, namely minimum values for object area, intensity and lifetime. We find that the tracking method has no impact on the detected climate-change signal. The criteria for identifying a convective object, however, can have a

strong and statistically significant impact on the magnitude of the climate-change signal, for all analysed object properties. For the case considered in our study, this insight reveals that projected changes in the characteristics of convective rainfall vary considerably between cells of differing intensity, area and lifetime; for example, an increase in the area of moderate-intensity cells alongside a decrease for the most intense cells. Our results suggest that for Lagrangian analysis of precipitation in climate models, sensitivity analysis of the climate-change signal in relation to how an object is defined is a useful enhancement.

## 1 Introduction

Lagrangian analysis of convective precipitation offers an alternative to the more common Eulerian approach, in which precipitation is considered at a fixed location. In the Lagrangian framework, often referred to as "cell", "storm", "feature" or "object-oriented" tracking, convective "objects" are identified and then tracked through space and time. The approach has historically been mostly used in radar-based nowcasting, in which the location of convective cells is forecast based on Lagrangian

advection from previous radar scans (Dixon and Wiener, 1993; Golding, 1998; Mandapaka et al., 2012; Novo et al., 2014). The Lagrangian approach furthermore allows the properties of convective objects to be measured during the object's life cycle. Characterizing these properties – e.g. area, mean or maximum intensity, distance travelled, etc. – has applications in both model evaluation and climate-change and impact studies. In the former, aspects of model-simulated convective precipitation which would not be discernible from Eulerian analysis – e.g. cell areal extent, lifetime, distance travelled, etc. – can be compared

with radar-based observations (Caine et al., 2013; Brisson et al., 2018; Purr et al., 2019; Caillaud et al., 2021; Raupach et al.,





2021), avoiding the double-penalty problem and potentially revealing previously unknown model strengths or weaknesses (Clark et al., 2014; Skinner et al., 2018). For climate-change studies, Lagrangian techniques can identify the relative changes in different storm properties, thus offering additional insight into the physical mechanisms underlying projected future changes in convective precipitation (Purr et al., 2021; Prein et al., 2017; Poujol et al., 2020a). For impact studies, multiple factors such

as storm motion, translation speed and spatiotemporal variability affect the drainage response of a catchment (Amengual et al., 2021): object-oriented analysis allows these factors to be quantified.

The many object-based algorithms used to track convective precipitation employ a number of different approaches, which include (i) pattern-matching-, (ii) overlap- and (iii) advection-based techniques, as well as combinations of the aforementioned. In pattern-matching approaches (Einfalt et al., 1990; Dixon and Wiener, 1993), the precipitation fields at successive time steps

are compared and object motions are determined based on spatial correlation or some other optimization method which matches objects with similar characteristics. With overlap-based methods (Morel and Senesi, 2002; Hering et al., 2004; Davis et al., 2006), the aim is to find object footprints which are contiguous in both space and time (i.e. spatial overlap at successive time steps). This approach may, in certain cases, be unsuitable for application with radar data: if the scans are too infrequent, contiguity will be lost; even in models, very small objects may also not overlap at successive timesteps. In the advection-based

approach, the expected position of the object is estimated based on Lagrangian extrapolation from the previous time step(s). Extrapolation may be based on, for example, mid-tropospheric flow (Purr et al., 2019; Brendel et al., 2014; Moseley et al., 2013), optical flow methods (He et al., 2019; Muñoz et al., 2018; Woo and Wong, 2017) or advection of some otherwise computed velocity field (Stein et al., 2014; Germann and Zawadzki, 2002). The approach may be unsuitable in situations of back-building (Parodi et al., 2017), where cold-pool outflows cause the convective system to propagate against the direction of

flow.

The desire to track convective objects naturally raises the question of what exactly is a convective object? How should it be defined? For tracking purposes, convective objects are typically defined based on exceedance of three threshold minima: (1) minimum precipitation intensity, (2) minimum area, and (3) minimum lifetime. Some tracking algorithms employ a fourth criterion, whereby precipitation must also be identifiable as convective, e.g. based on cloud-top temperatures (Chen et al.,

2019), precipitation gradients (Brendel et al., 2014) or mid-tropospheric dynamics (Poujol et al., 2020b). The choices of the aforementioned thresholds vary considerably in the literature: minimum intensities from 0.1 mm h$^{-1}$ (Li et al., 2020) to 30 mm h$^{-1}$ (Caillaud et al., 2021); area thresholds as low as 2 or 4 km$^2$ (Moseley et al., 2013; Stein et al., 2014) and as high as 32,000 km$^2$ (Prein et al., 2017); time thresholds of 10 min (Moseley et al., 2013), 30 min (Burghardt et al., 2014) or even longer in low temporal-resolution data (Li et al., 2020). While it seems obvious that the choice of how to define a convective

object will impact the climatological statistics of certain object properties, what is not clear is if these choices may also impact the climate-change response of convective objects' characteristics. The same question may also be posed of the chosen tracking method.

To investigate these questions, we employ the pseudo global warming (PGW) method (Schär et al., 1996) to perform high-resolution ensemble climate-change simulations with a convection-permitting model (CPM). Our PGW ensemble covers a

two-week period of exceptionally high thunderstorm activity over central Europe (Piper et al., 2016). CPMs offer an ideal tool





to investigate such questions, as they explicitly represent deep convection. In our study region, CPMs have been shown to add value for the representation of both the diurnal convective cycle (Meredith et al., 2021; Brisson et al., 2016b) and intense convective precipitation (Fosser et al., 2015; Knist et al., 2018). Importantly for the tracking of convective objects, CPMs – here, the COSMO-CLM (Rockel et al., 2008) – can realistically represent many aspects of subhourly precipitation from both Eulerian (Meredith et al., 2020) and Lagrangian (Brisson et al., 2018; Purr et al., 2019) perspectives. Using two different tracking methods, based on the overlap and advection approaches, we track all convective objects in the aforementioned (present and future) PGW ensemble. The tracking is repeated using different options for defining a convective object: the minimum intensity, area and lifetime thresholds discussed above are systematically varied. The aim is to see how sensitive the warming response of different object characteristics is to the chosen tracking method and the manner in which a convective object is defined.

In the first results section (Section 4), our purpose is to pose the question: in the presence of a climate change signal, can projected changes in the characteristics of convective cells be sensitive to the choice of tracking algorithm or to how a convective object is defined? We are thus interested in *differences* in the climate-change signal, rather than precisely determining the magnitude of convective objects' response to climate change in our region. In the second results section (Section 5), we asses the magnitude of the projected changes and – based on any sensitivities identified in the preceding section – explore how Lagrangian projections might be analysed so that projections are less sensitive to how a convective object is defined.

## 2 Study period

Our study makes use of a two-week period of unusually high convective activity over Germany, 26[th] May to 9[th] June 2016 and analysed in detail in Piper et al. (2016). The exceptional number of thunderstorms over an extended period led to flash flooding and serious structural damage in many locations (e.g. Bronstert et al., 2018). The study period can be roughly split into two parts: a first part in which convection was caused by a strong synoptic forcing (Fig. 1a), and a second in which weak forcing (Fig. 1b) gave rise to a daily cycle of instability building over large areas, followed by intense convection in the late-afternoon and evening (Hirt and Craig, 2021). Owing to its elevated levels of both weakly- and strongly-forced convection, the period has previously been used as a test case in numerous studies of convection in kilometre-scale models (Baur et al., 2018; Rasp et al., 2018; Keil et al., 2019; Hirt et al., 2019; Hirt and Craig, 2021). The period of strong synoptic forcing included south-easterly advection of warm and moist air into Germany, large-scale uplift from a strong potential vorticity anomaly on the 29[th], followed by a number of near-stationary surface lows over central Europe under a 500 hPa cut-off low. The weakly-forced convection occurred under an upper-level stationary ridge. Further discussion is available in the aforementioned references.



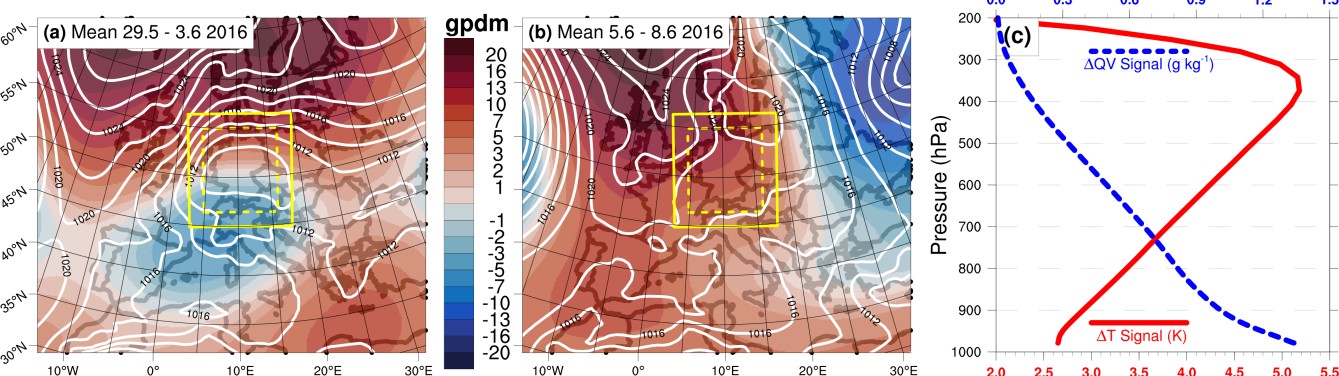

**Figure 1.** 500 hPa geopotential height anomaly (shading; reference is 1979-2015 mean) and sea level pressure (white lines) averaged over the periods (a) 29.5–3.6 and (b) 5.6–8.6, 2016. The maps cover the spatial extent of the 0.11° simulation domain and the solid and dashed yellow lines mark the 0.025° simulation domain and analysis region, respectively. (c) Climate-change signal of temperature and specific humidity added to the initial and boundary conditions of the 0.11° PGW simulations. The signal is computed based on an area average over the 0.11° domain.

## 3 Methods

### 3.1 Climate simulations

We perform 18-member ensemble regional climate model (RCM) simulations of our study period using the PGW approach (Schär et al., 1996) at convection-permitting resolution (0.025°, ∼2.8 km). In the PGW approach, an event or period is first dynamically downscaled from reanalysis under present conditions. The downscaling is then repeated with altered RCM initial and boundary conditions which reflect projected changes in the boundary variables (or a subset of the variables). This approach has previously been employed in numerous studies on both climate and event-based timescales (Prein et al., 2017; Lackmann, 2013; Rasmussen et al., 2014; Kröner et al., 2017; Keller et al., 2018; Hibino et al., 2018). All of our simulations are performed with the COSMO-CLM (Rockel et al., 2008), version 5.0_clm16.

The first modelling step (present climate) involves multi-reanalysis downscaling of ERA-Interim (Dee et al., 2011) and MERRA2 (Gelaro et al., 2017) to 0.11° resolution from 26.05.2016 to 09.06.2016 over a pan-Europe domain (Fig. 1). An ensemble is then created using the domain-shift technique (e.g. Rezacova et al., 2009; Pardowitz et al., 2016; Noyelle et al., 2018). In this approach, a central domain is defined and the domain centre is systematically shifted five grid cells (∼0.55°) in the cardinal and ordinal directions N, NE, E, SE, S, SW, W and NW, giving perturbed initial and boundary conditions for each ensemble member (see Rezacova et al. (2009) and Mazza et al. (2017) for illustrative schematics). The shifting is performed for both reanalyses, giving in total 18 members for the present climate.

The second modelling step (PGW) involves repeating the 0.11° downscaling with modified boundary conditions based on projected changes under the RCP8.5 scenario (Van Vuuren et al., 2011), as described above. We derive an ensemble mean climate-change signal from historical (1970-1999) and future (2070-2099) periods based on three 0.11° COSMO-CLM sim-



ulations from the EURO-CORDEX experiment (Jacob et al., 2014). The three EURO-COREX runs were downscaled from CMIP5 (Taylor et al., 2012) simulations of the MPI-ESM-LR (r1; Giorgetta et al., 2013), EC-EARTH (r12; Hazeleger et al., 2012) and CNRM-CM5 (r1; Voldoire et al., 2013) global models. A 31-day running mean of the resulting climate-change signal (Fig. 1c) is added to the initial and lateral boundary conditions of our 0.11° simulations for all variables (e.g. temperature, specific humidity, pressure and winds).

Finally, all present and PGW members are further downscaled to 0.025° resolution over the geographically fixed (i.e. not shifted) COSMO-DE domain (Fig. 1); this gives an 18-member CPM ensemble from 27.05.2016 to 09.06.2016 (14 days). Deep convection is explicitly resolved by the model, while shallow convection is parametrized based on a modified Tiedtke scheme (Tiedtke, 1989). All model settings are taken from the standard configuration of the German Weather Service and precipitation output is saved every 5 min. Aside from the added value of the COSMO-CLM, and CPMs in general, discussed in the introduction, shortcomings in the COSMO-CLM do still remain. Keil et al. (2014) reported insufficient convective triggering under conditions of weak synoptic forcing, while Purr et al. (2019) reported an underestimation of mean precipitation intensity in long-living, extreme convective objects and a general overestimation of the lifetime of convective objects. The results presented below are all based on the 0.025° CPM ensemble.

## 3.2  Tracking algorithms

We make use of two tracking algorithms. In the first, convective objects are tracked based on advection by the steering flow; we refer to this algorithm as ADV. In the second, convective objects are tracked based on the overlap method; we refer to this algorithm as OVER. These algorithms are chosen (i) because they are representative of two standard approaches to tracking convective objects (i.e. advection- and overlap-based tracking), and (ii) for their low levels of complexity, facilitating generalizability of the results.

The ADV algorithm is based on the method of Brendel et al. (2014), which was developed for tracking convective objects in radar data and was adapted for convection-permitting models by Brisson et al. (2018). The OVER algorithm, on the other hand, is a simple temporal overlap procedure. The algorithms have been summarised in a schematic (Fig. 2). For both algorithms, non-convective precipitation is first masked out using the method of Poujol et al. (2020b). All precipitation below a chosen threshold ($P_{min}$) is also masked out. Objects are then identified as contiguous precipitation areas exceeding a minimum chosen area ($A_{min}$), based on the number of grid boxes within the object. Objects whose lifetime is shorter than a chosen threshold ($T_{min}$) are discarded, as are objects which are not fully in the domain.

In ADV (Fig. 2a), once an object has been identified, its position at the next timestep is estimated based on the steering flow, here the wind velocity averaged across the 500, 700, and 850 hPa levels. From the expected location at the next timestep, convective objects are searched within a defined search radius whose length is proportional to the wind speed (see Brendel et al. (2014)). For the object nearest to the expected location, the procedure is further iterated until no object is found. For object splits, the object nearest to the expected location is chosen, while the remaining object(s) is (are) considered as a new object(s). For object mergers, the largest of the original objects is continued, while the other track is ended.





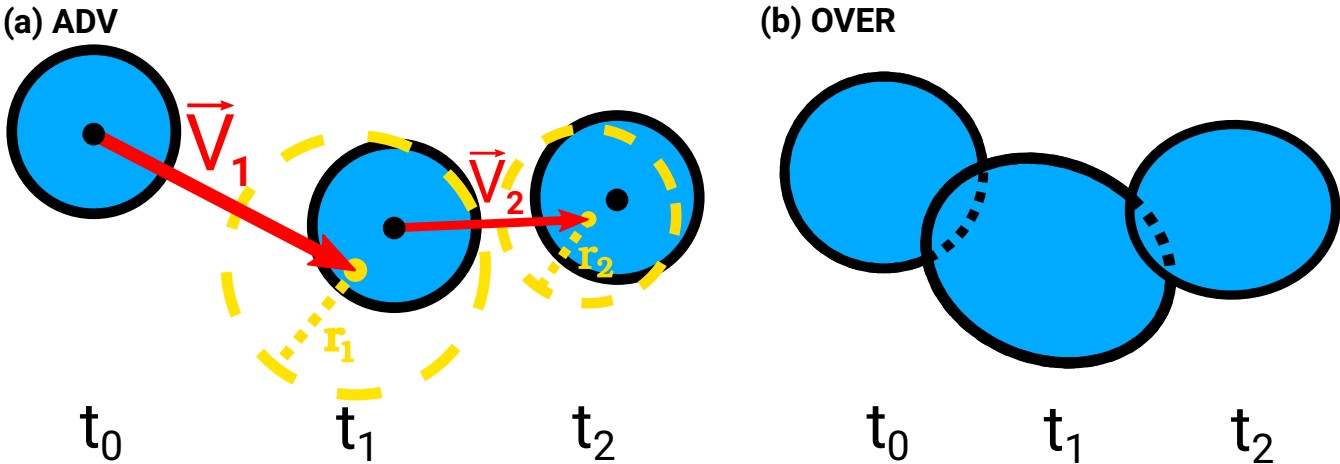

**Figure 2.** Schematic illustrating the (a) ADV and (b) OVER algorithms. In ADV, the red vectors (emanating from the cell's centre of mass) represent the estimated displacement of the convective cell based on the steering flow. The green dashed circle represents the search area, with radius **r**, in which the displaced cell is sought. The search radius is proportional to the magnitude of the displacement vector. In OVER, the area between the dashed and solid lines marks the cell's area-of-overlap at consecutive timesteps. See also Brisson et al. (2018) for a schematic of the Brendel et al. (2014) algorithm.

In OVER (Fig. 2b), the spatial footprint of an identified object is first determined and an overlap between this footprint and any footprints at the next timestep is sought. The process is further iterated until no overlap is found. For both object splits and mergers, the object with the largest overlap (by precipitation volume) is continued, while the other object is considered new (splits) or to have ended (mergers).

145    Both algorithms compute the following lifetime diagnostics for each object: mean and maximum areal precipitation intensity ($P_{avg}$, $P_{max}$), mean and maximum object area ($A_{avg}$, $A_{max}$), mean and maximum integrated precipitation volume ($Vol_{avg}$, $Vol_{max}$), lifetime ($T$), total distance travelled ($D$) and average speed ($S$). We use 5 min precipitation totals in our study.

### 3.3    Analysis

The ensemble setup of 14-day CPM simulations provides an ideal platform to test a wide range of options for defining a
150    convective object and comparing two tracking algorithms. The aim is to see whether, in the presence of climate warming, the tracking algorithm or how a convective object is defined may impact the magnitude of any detected changes in the characteristics of convective objects. As mentioned in Section 3.2, we analyse the object characteristics $P_{avg}$, $P_{max}$, $A_{avg}$, $A_{max}$, $Vol_{avg}$, $Vol_{max}$, $T$, $S$ and $D$ over the lifetime of each object. For each ensemble member, we consider the median of these metrics; the change of the associated ensemble mean is then computed. In addition to the characteristics of convective objects,
155    we also consider the total number of convective objects ($N_{obj}$) and the total volume of convective precipitation ($P_{tot}$). Our analysis region is removed from the boundaries of the 0.025° simulation domain (Figs. 1, 3) in order to allow sufficient spinup of convective features (Brisson et al., 2016a).





The options for defining a convective object which we vary are as follows. (1) The minimum area of the object $A_{min}$, which is based on the number of grid boxes ($N$) within the object. Each grid box has an area of $\sim$7.7 km$^2$ and we consider values for $A_{min} = 2^i$ grid boxes, where $i = 1 \ldots 6$. (2) The minimum precipitation threshold for the object ($P_{min}$), with equivalent hourly rates of 4.5, 6.5, 8.5, 10.5 and 12.5 mm h$^{-1}$ considered. (3) The minimum lifetime of the object ($T_{min}$), with thresholds of 15, 30, 45, 60, 90 and 120 min considered. These options are varied across both tracking algorithms, giving a total of 132 different tracking configurations. To aid comparison, we also define a reference setup, with $A_{min} = 8$ grid boxes, $P_{min} = 8.5$ mm h$^{-1}$ and $T_{min} =$15 min. These reference settings are chosen based on a balance between previous studies in our region (Brisson et al., 2018; Purr et al., 2019, 2021) and finding a mid range between the tested options. Results for the reference settings are presented in Table 1, where the absolute values can also be found.

### 3.4 Uncertainty and significance

To test and conveniently display the statistical significance of any differences in the detected change signals, we employ bootstrap resampling in conjunction with the confidence intervals (CIs) proposed by Goldstein and Healy (1995). All ensemble members are first resampled 10,000 times with replacement and the change signal is re-computed each time, giving a distribution of 10,000 changes. Under the normal approximation, the bootstrap CIs for the statistic $t_i$ can be constructed as $\theta_{i,\alpha/2}, \theta_{i,1-\alpha/2} = t_i \pm z_\alpha \sigma$ (Davison and Hinkley, 1997), where $\alpha$ is the two-tailed probability, $z_\alpha$ the corresponding positive gaussian quantile, and $\sigma$ the standard deviation. In the case of two change statistics $t_i$ and $t_j$, their differences will be statistically significant at level $\alpha$ if the condition $|t_i - t_j|/\sqrt{\sigma_i^2 + \sigma_j^2} > z_\alpha$ is satisfied. Their CIs, meanwhile, will be non-overlapping if $|t_i - t_j|/(\sigma_i + \sigma_j) > z_\alpha$. Rewriting the left-hand side of the latter in terms of the former, it can be shown that differences significant at level $\alpha$ will have non-overlapping CIs constructed as

$$\theta_{i|j,\beta/2}, \theta_{i|j,1-\beta/2} = t_{i|j} \pm z_\beta \sigma_{i|j} \,, \tag{1}$$

where

$$z_\beta = z_\alpha \frac{\sqrt{\sigma_i^2 + \sigma_j^2}}{\sigma_i + \sigma_j} \,. \tag{2}$$

This can be repeated across multiple categories to compute a single $z_\beta$, which is the average taken across all pairs $i, j$; each category $i \in \mathbb{Z}^+$ then has CIs $t_i \pm z_\beta \sigma_i$ (Goldstein and Healy, 1995). Statistically significant differences between the different change signals can hence easily be discerned from an absence of overlap between the Goldstein-Healy CIs. In our study, we take $\alpha = 0.95$.

## 4 Results I: Sensitivity of climate-change signal

### 4.1 Reference setup

We begin with a reference setup for both algorithms (ADV and OVER): a minimum area $A_{min} = 8$ grid boxes, a minimum precipitation threshold $P_{min} = 8.5 \ mm \ h^{-1}$ (0.7 $mm/5 \ min$) and a minimum lifetime $T_{min} = 15 \ min$. This setup serves as



**Table 1.** Present, future and relative change values of $N_{obj}$ and all object properties, for the ADV and OVER tracking algorithms. The results are based on the reference setup (Section 4.1): $A_{min} = 8$ grid boxes, $P_{min} = 8.5$ mm h$^{-1}$, $T_{min} = 15$ min. For display purposes, the table entries have been rounded, which explains any slight deviations of the relative changes from that expected based on the present and future entries. Brackets denote confidence intervals, computed as described in Section 3.4.

|  | Present | | Future | | Change (%) | |
|---|---|---|---|---|---|---|
|  | ADV | OVER | ADV | OVER | ADV | OVER |
| $N_{obj}$ | 4,620 | 4,594 | 6,785 | 6,745 | +46.9 [42.0,51.7] | +46.8 [41.9,51.7] |
| $L$ (min) | 35.0 | 35.0 | 35.3 | 35.6 | +0.8 [-0.2,1.9] | +1.6 [0.0,3.2] |
| $D$ (km) | 11.6 | 11.7 | 15.9 | 16.1 | +37.2 [34.8,39.6] | +37.8 [35.6,40.0] |
| $S$ (m s$^{-1}$) | 5.9 | 5.9 | 7.7 | 7.7 | +30.4 [27.8,32.9] | +30.5 [27.8,33.1] |
| $A_{avg}$ (km$^2$) | 95.4 | 95.1 | 102.1 | 101.7 | +6.9 [6.7,9.0] | +6.9 [5.9,7.8] |
| $A_{max}$ (km$^2$) | 125.6 | 125.2 | 136.9 | 136.9 | +8.9 [7.6,10.2] | +9.3 [7.8,10.8] |
| $P_{avg}$ (mm h$^{-1}$) | 18.0 | 18.1 | 17.0 | 17.0 | -5.6 [-6.7,-4.6] | -5.6 [-6.7,-4.5] |
| $P_{max}$ (mm h$^{-1}$) | 45.3 | 45.5 | 43.1 | 43.4 | -4.8 [-6.6,-3.0] | -4.6 [-6.4,-2.8] |
| $V_{avg}$ ($10^5$ m$^3$) | 1.45 | 1.44 | 1.50 | 1.50 | +3.8 [2.5,5.0] | +3.9 [2.7,5.1] |
| $V_{max}$ ($10^5$ m$^3$) | 1.94 | 1.93 | 21.0 | 21.0 | +8.5 [6.9,10.1] | +8.6 [7.0,10.2] |
| $P_{tot}$ ($10^{10}$ m$^3$) | 2.01 | 1.86 | 3.75 | 3.68 | +87 [72,102] | +98 [87,110] |

a threshold "base-state" at which in the following sections at least one threshold ($A_{min}$, $P_{min}$, $T_{min}$) is held constant while the remaining threshold(s) vary singularly or jointly. Under this setup (Table 1), we find ensemble medians of about 4,500 objects

per member, which are concentrated in the western half of the analysis region (Fig. 3). Median lifetimes and distances travelled for the objects are roughly 35 min and 12 km, respectively, for each algorithm. For the lifetime object mean precipitation rates and areas, an equivalent hourly rate of 18 mm h$^{-1}$ and an area of 96 km$^2$ are found. In the PGW ensemble, the total numbers of objects increases by over 45%. Changes in the object characteristics in response to the PGW signal range from -6 % to +38 % (Table 1), depending on the object characteristic. The greatest increase is seen in distance travelled, with minimal

change in object lifetime. Object areas and volumes increase, with areal mean precipitation intensity decreasing. The net effect of the aforementioned changes on total convective precipitation is an increase of roughly 87 (ADV) to 98 (OVER) %, which is the most noticeable difference between the two tracking methods. Amongst all change signals, no statistically significant differences between ADV and OVER are evident.

## 4.2 Minimum size of object ($A_{min}$)

In this subsection, we hold $T_{min}$ and $P_{min}$ constant at their reference values. $A_{min}$ (the minimum area threshold) is varied, with values of $A_{min} = 2^i$ grid boxes, where $i = 1 \ldots 6$ (Fig. 4). For $P_{avg}$ and $P_{max}$ (the object lifetime mean and maximum precipitation intensity), the minimum object size has no significant impact on the response to warming; this is mostly true for the object lifetime too. For the remaining metrics, however, the $A_{min}$ threshold has a significant impact on the resulting climate-



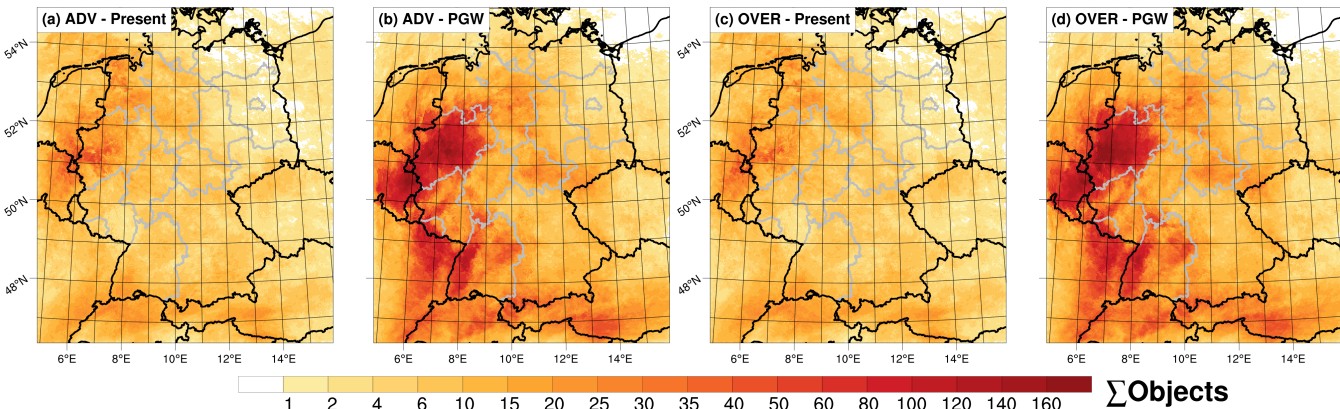

**Figure 3.** Total number of objects counted at each grid box for ADV in the (a) present and (b) PGW ensembles, and OVER in the (c) present and (d) PGW ensembles. Results are based on the algorithms' reference setup. The analysis region is as denoted by the dashed yellow boxes in Fig. 1. Note that a higher number of objects does not necessarily correspond to higher precipitation, e.g. one large system could cause more precipitation than multiple smaller cells.

change signal. For volume ($V_{avg}$, $V_{max}$), area ($A_{avg}$, $A_{max}$), distance travelled and average speed of the objects, the strongest

climate-change signal is found for the lowest $A_{min}$, with the weakest signal for the highest $A_{min}$. For the aforementioned object characteristics, the response to warming using the lowest $A_{min}$ threshold (2 grid boxes) is an order of magnitude greater than with the greatest $A_{min}$ threshold (64 grid boxes). Right across the different $A_{min}$ thresholds tested, statistically significant differences in the magnitude of the climate-change signal are found (as evident from the non-overlapping Goldstein-Healy CIs; see Methods). For the number of convective objects ($N_{obj}$), the trend is reversed: the higher the $A_{min}$ threshold, the stronger

the climate-change signal, again with statistically significant differences. The different tracking methods are found to have no statistically significant difference in their computed climate-change signals.

An important point to note is that depending on the chosen $A_{min}$ threshold, the physical interpretation for why total convective precipitation increases in the warmer climate (Fig. 7) could be different. For small $A_{min}$, the increase in convective precipitation would appear to be driven by the area and volume of the objects increasing. For larger $A_{min}$, on the other hand,

the increase in total precipitation would appear to be driven by strong growth in the number of convective objects. That is not to say the one choice of $A_{min}$ is "wrong" or another "correct", but rather to recognize that the role of object characteristics in changing total convective precipitation is conditional on how a convective object is defined, and that results should be interpreted in this context. These differences are worth bearing in mind when drawing inferences about future changes in the characteristics of convective precipitation.

**4.3   Minimum precipitation intensity of object ($P_{min}$)**

In this subsection, $A_{min}$ and $T_{min}$ are fixed at their reference values, while the precipitation-minimum threshold $P_{min}$ is varied across values of 4.5, 6.5, 8.5, 10.5 and 12.5 mm h$^{-1}$ (using the equivalent 5-min rate). The choice of $P_{min}$ threshold has





**Figure 4.** Climate-change signals of different object properties as a function of the object's **minimum-area criterion** $A_{min}$, for both algorithms. Change signals which are different with statistical significance at the 0.95 level can be identified based on non-overlapping CIs (Section 3.4). In panel (a), the absolute values for the number of objects are shown (i.e. the sample sizes). $A_{min}$ is defined in terms of grid boxes, with each grid box having an area of ~7.7 km$^2$. The $A_{min}$ range thus spans approximately 15 to 493 km$^2$. The absolute values underlying the change signals can be seen in Fig. S1.

much less of an impact on the magnitude of the climate-change signal than varying $A_{min}$. Across the sampled range of $P_{min}$ thresholds, clear statistically significant differences (Fig. 5) are most evident for diagnostics which characterize the object's precipitation intensity: $P_{avg}$ and $P_{max}$ show a monotonic upward trend in their climate-change signal with increasing $P_{min}$; this is in contrast to varying $A_{min}$, which was shown to have no effect on the climate-change signals of $P_{min}$ and $P_{max}$. Some smaller but statistically significant differences are also seen in the object area's response to warming ($A_{avg}$, $A_{max}$; Fig. S2) and in the total number of objects. For the remaining object characteristics, the range of tested $P_{min}$ thresholds produces very few significant differences in the response to warming. The speed of the objects does, however, show a clear monotonically






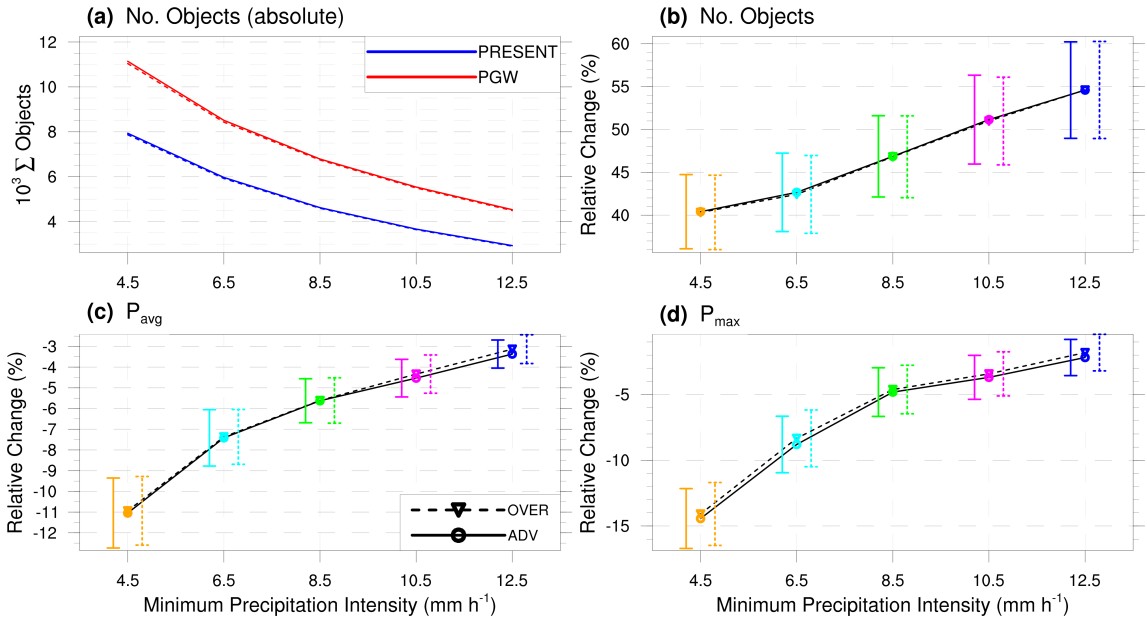

**Figure 5.** Climate-change signals of $P_{avg}$, $P_{max}$ and $N_{obj}$ as a function of the object's **minimum-precipitation-intensity criterion** $P_{min}$, for both algorithms. Change signals which are different with statistical significance at the 0.95 level can be identified based on non-overlapping CIs (Section 3.4). In panel (a), the absolute values for the number of objects are shown (i.e. the sample sizes). $P_{min}$ is shown as the equivalent hourly rate based on 5 min intensities. The climate-change signals of the remaining object properties, and the absolute values underlying them, can be seen in Figs. S2 and S3.

decreasing trend (Fig. S2), suggesting that over a wider range of $P_{min}$ thresholds, significant differences may emerge. As with the $A_{min}$ threshold, no statistically significant differences between the tracking methods are evident.

### 4.4 Minimum lifetime of object ($T_{min}$)

Here we vary the minimum-lifetime threshold $T_{min}$ of the objects, while keeping $P_{min}$ and $A_{min}$ at their reference values (Fig. 6b,e); this is then additionally shown for the smallest and largest values of $A_{min}$ (2 and 64 grid boxes; see Fig. S4 for

remaining $A_{min}$ values). Starting with the reference values of $P_{min}$ and $A_{min}$, it is found that varying the minimum-lifetime threshold $T_{min}$ has a clear and statistically significant impact on the magnitudes of the climate-change signals of the speed, distance travelled and lifetime object characteristics in both algorithms, as well as for the total number of objects. To a lesser extent, significant differences are found for the area metrics.

Looking at the smallest and greatest values of $A_{min}$, it is only the speed, distance travelled and lifetime properties which

consistently display climate-change signals that are sensitive to how an object's minimum lifetime ($T_{min}$) is defined. At the smallest $A_{min}$ threshold ($A_{min} = 2$), all diagnostics are found to exhibit a climate-change signal with some degree of sensitivity to the magnitude of $T_{min}$, with the strongest sensitivities for the aforementioned properties, as well as the number of





objects. As $A_{min}$ increases, the impact of $T_{min}$ on the magnitude of the climate-change signal generally decreases and is either eliminated or greatly reduced by the maximum ($A_{min} =64$ grid boxes; see also Fig. S4). Comparing the two tracking methods,

no statistically significant differences are found between the algorithms.

## 4.5  Total convective precipitation

Changes in the characteristics of convective objects do not necessarily inform us about changes in total convective precipitation. An additional metric of interest in object-oriented precipitation analysis may thus be the total amount of convective precipitation attributable to the identified objects ($P_{tot}$), and how this responds to warming. By jointly varying (i) $T_{min}$ and $A_{min}$, and

(ii) $T_{min}$ and $P_{min}$, a large range of $P_{tot}$ responses is found across 132 setups, with a strong $P_{tot}$ increase in all cases (Fig. 7), ranging from about +70 % to +120 %, depending on the combination of the three thresholds. As with the reference setup (Section 4.1), considerable differences are often evident between the two algorithms, with those for OVER typically stronger. However, due to the large range of uncertainty in the magnitude of these increases, no statistically significant differences between the tracking methods are found.

A general, though not uniform, pattern of a stronger warming response with higher $T_{min}$ thresholds and lower $P_{min}$ thresholds can be discerned, while no clear influence of the $A_{min}$ threshold on the $P_{tot}$ climate-change signal is evident. The higher increases in total precipitation with higher $T_{min}$ thresholds mirror the changes seen for the number of objects as $T_{min}$ increases (Fig. 6), suggesting that the latter explains differences in the climate-change signal of $P_{tot}$ as $T_{min}$ is varied. Higher increases in $P_{tot}$ as $P_{min}$ decreases, meanwhile, appear to be explained by differences in the $A_{avg}$ signal as $P_{min}$ is varied (Fig. S2f).

## 5  Results II: Future projections

In this second results section, projected future changes to cell characteristics based on our PGW simulations are evaluated. Here, we consider the magnitude of the projected changes, as opposed to differences in the magnitude under the different thresholds presented in previous sections. We consider how Lagrangian projections might best be presented based on the lessons learned in Sections 4.2 to 4.5 and compare the projections with those found in the literature for our region. As it has

been shown in previous sections that the choice of tracking method has no impact on our results, we will for clarity show results for just the ADV algorithm.

Purr et al. (2021) performed Lagrangian analysis of convective cells over Germany under present and future (RCP8.5) conditions using continuous 30-year simulations. Their model and resolution were the same as in this study (COSMO-CLM, 0.025°) and thresholds of $A_{min} = 4$ grid boxes, $T_{min} = 15$ min and $P_{min} = 8.5$ mm h$^{-1}$ were used. Under these thresholds

and accounting for uncertainties, our results for $T$ are in agreement, whereas our projected changes for $A_{max}$ and $S$ are higher and for $P_{avg}$ and $P_{max}$ are lower (our other metrics weren't considered in the study). This confirms that our 14-day study period of high convective activity is not representative of climatological conditions; our projections should thus be considered as indicative for the synoptic conditions present during our study period.





In Sections 4.2 to 4.5 it was shown that the choice of thresholds for defining a convective object can significantly impact
the magnitude of the climate-change signal. We therefore propose analysing the output of the tracking algorithm by first
partitioning the data into bins delineated by different values of $A_{avg}$, $P_{avg}$ and $T$, the metrics on which the object thresholds are
based. To maximize the range covered by all bins, the trackings with each of the three lowest thresholds – $A_{min} = 2$ grid boxes,
$P_{min} = 4.5$ mm h$^{-1}$ and $T_{min} = 15$ min – alongside their counterpart reference thresholds are used (Fig. 8).

Considering all metrics, it is found that, in the RCP8.5 scenario, the greatest projected increase is for the number of convec-
tive objects ($N_{obj}$), which has maxima for (i) low- and high-intensity objects (close to a doubling), and objects with (ii) large
area and (iii) long lifetimes. Partitioning objects based on their intensity reveals a strong increase in most metrics across the
spectrum (Fig. 8a), which reaches a maximum for objects with mid-range intensities ($\sim 9 - 15$ mm h$^{-1}$): future increases of
$10 - 20$ % for object area and volume, or as high as 60 % for the distance an object travels. Using object area as a basis for
partitioning (Fig. 8b), meanwhile, a general negative change for most object characteristics is found: decreases of $5 - 20$ % in
volume and intensity for the smallest objects. Exceptions to this are for distance travelled and speed, where increases of over
50 % are projected in some bins. One thing that all object characteristics have in common is that as object area increases, their
projected change tends asymptotically to a common value. This mirrors many of the results shown in Figs. $4 - 6$ and suggests
that the spatial homogeneity of the precipitation field is an important factor in the sensitivity of Lagrangian projections to
object thresholds, i.e. larger area thresholds ($A_{min}$) give projections whose magnitude is less sensitive to further increases in
the area threshold. Similar to the $T_{min}$ thresholds presented in Fig. 6b, the object lifetime only has a clear impact on future
changes in objects' distance travelled and speed (Fig. 8c), ranging from 50 % for the shortest-lived objects (15 min) to 10 %
for the longest-lived objects (> 3 h). Meanwhile, the other object characteristics show future changes which vary little across
the spectrum of object lifetimes, with projected changes mostly within $\pm 10$ %.

The results following from the above partitioning (Fig. 8) may appear somewhat contradictory. For example, a generally
positive change in characteristics when partitioning by object intensity (a) and a generally negative change in most character-
istics when partitioning based on object area (b). Or, additionally, a negative change in volume when partitioning by object
area (b) and a positive change in volume when partitioning by object lifetime (c). In Fig. 7 it has already been shown that total
precipitation increases in all cases. The increase in $P_{tot}$ can be explained – without having to test a myriad of object thresholds
– by the insights revealed from the partitioning applied in Fig. 8. From here, the following picture emerges for our study period:
(1) the total number of objects increases in (almost) all cases, (2) future objects have larger areas and volumes, regardless of
how long they live or how intense they are, (3) despite this, objects of all areas and lifetimes have lower mean intensities, (4) it
can thus be concluded that the increase in $P_{tot}$ is driven by the combined effect of more objects and an increase in the area
of these objects, (5) the increase in object volumes despite a decrease in intensity shows that the effects of more objects and
higher areas are dominant over the reduction in mean object intensity, which acts in the opposing direction. A final note would
be that the area of the most intense objects is actually found to decrease (-6 %) and their maximum local precipitation intensity
found to increase (+12 %), in agreement with the findings of Armon et al. (2022) and Caldas-Alvarez et al. (2022).





**Figure 6.** Climate-change signals of all object properties and $N_{obj}$, as a function of the object's **minimum-lifetime criterion** $T_{min}$, for both algorithms. Statistically significant difference in the climate-change signal against other $T_{min}$ thresholds is indicated by a number in the box centre. The number denotes how many of the other $T_{min}$ thresholds have a change signal whose difference is statistically significant compared to the box in question (maximum = 5). For example, for the combination (d) OVER, $A_{min} = 2$, Speed., $T_{min} = 15$ min, the number 3 is present: this means that the climate-change signal for this combination has a statistically significant difference to 3 of the remaining 5 $T_{min}$ thresholds of OVER, $A_{min} = 2$, Speed. Confidence intervals, computed as in Section 3.4, are given in the left-hand corners of each box. There are no statistically significant difference between the algorithms. The results are shown for three values of $A_{min}$, with the remaining values of $A_{min}$ found in Fig. S4 and absolute values in Fig. S5.



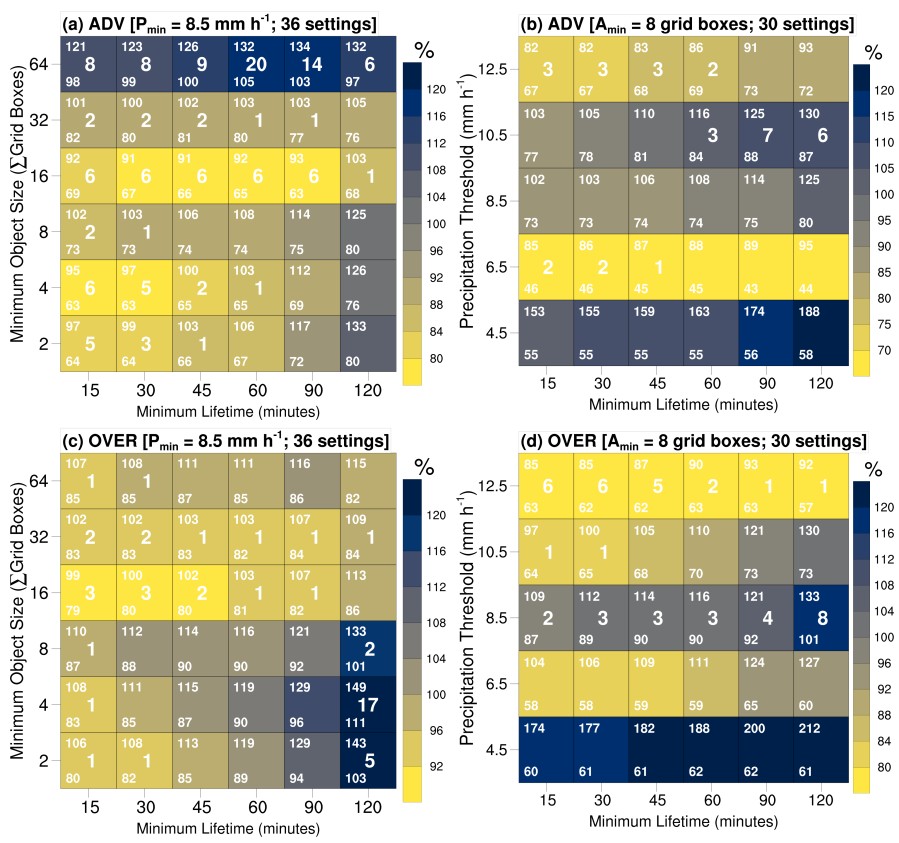

**Figure 7.** Change (%) in total convective precipitation in response to warming signal, for both algorithms. The change is based on the total precipitation attributable to all identified objects. In (a) and (c), $A_{min}$ and $T_{min}$ are jointly varied, with $P_{min}$ at its reference value. In (b) and (d), $P_{min}$ and $T_{min}$ are jointly varied, with $A_{min}$ at its reference value. Confidence intervals, computed as in Section 3.4, are given in the left-hand corners of each box. Statistically significant differences are denoted by a number in the middle of each square, as in Fig. 6. For example, for the case (c) OVER, $A_{min}$ = 2 grid boxes, $T_{min}$ = 120 min, the number 5 is present. This means that the change signal for this combination has a statistically significant difference to 5 of the remaining 35 configurations in (c). Note different colour-bar minima.



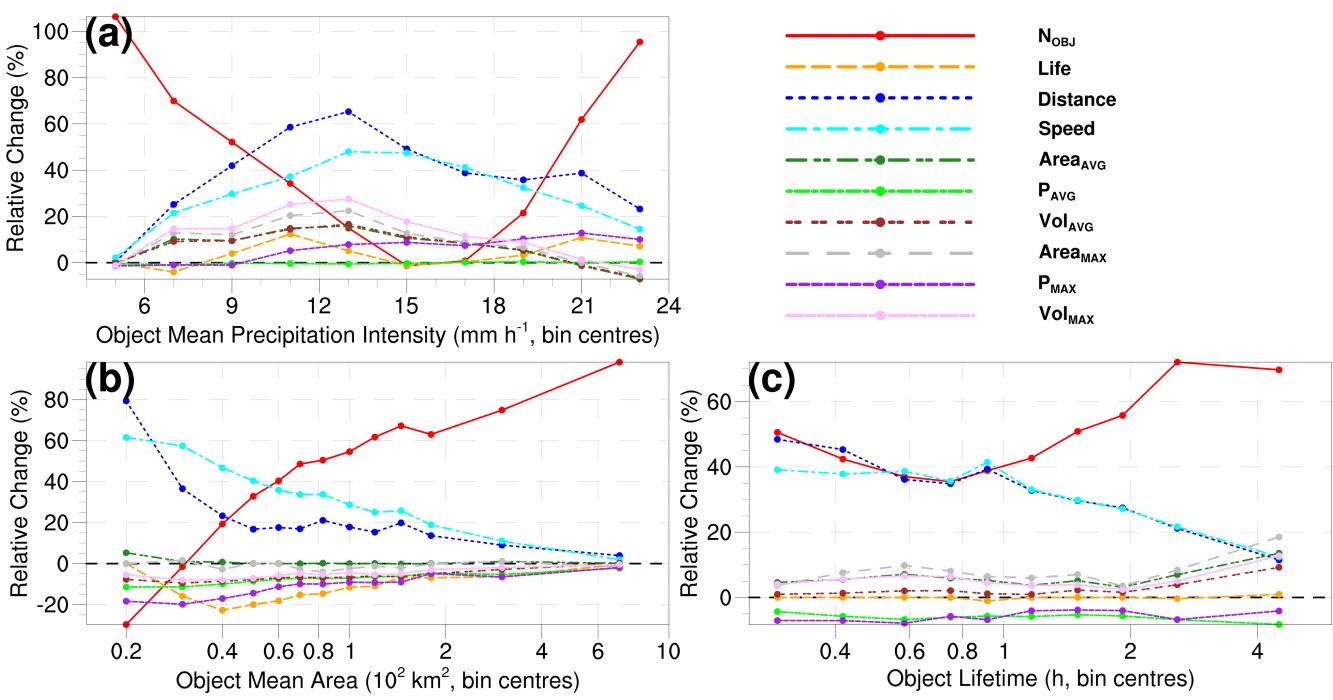

**Figure 8.** Ensemble mean projected change (%) in object characteristics under the RCP8.5 scenario as a function of (a) object intensity, (b) object area, and (c) object lifetime. Note the logarithmic x-axes in panels (b) and (c). In (a), (b) and (c), $P_{min}$, $A_{min}$ and $T_{min}$ are set at 4.5 mm h$^{-1}$, 2 grid boxes and 15 min, respectively, while the remaining thresholds are in each case set to their reference values. For visual clarity, results are based solely on the ADV algorithm. As in the rest of the manuscript, results are for median values. Each bin has a minimum of 50 data points in at least 15 of the 18 ensemble members; members with less than this total are not considered in the calculation.





## 6   Summary and Conclusions

Aided by the growing use of kilometre-scale climate models (Lucas-Picher et al., 2021), Lagrangian methods for analysing the response of convective precipitation to climate change have become increasingly popular (e.g. Prein et al., 2017; Poujol et al., 2020a; Purr et al., 2021). This object-oriented approach is particularly useful for studying changes in the characteristics of convective cells. In our study, we have tested the sensitivity of Lagrangian projections to the choice of (i) tracking algorithm and (ii) how a convective object is defined. Two simple tracking algorithms, each representative of a common approach to Lagrangian analysis, were employed to track convective objects in convection-permitting PGW ensemble simulations, allowing their respective climate-change signals to be compared. Furthermore, for each algorithm, the sensitivity of the climate-change signal to how a convective object is defined was examined by systematically varying the threshold criteria for identifying a convective object, namely: minimum size ($A_{min}$), intensity ($P_{min}$) and lifetime ($T_{min}$). In total, 132 configurations were tested. Our PGW simulations encompassed a 14-day period with elevated levels of both strongly- and weakly-forced convection (Section 2), offering a diverse representation of convective objects against which the different algorithms and configurations could be tested.

Our first main result is that the tracking method appears to have no significant impact on how the properties of convective objects, or the total number of convective objects, respond to climate change. The representative advection- and overlap-based algorithms which we implemented produce very similar climate-change signals for all object properties, with no statistically significant differences found. Additional tests of this conclusion using a set of climate-length simulations, those used in Meredith et al. (2019), show that the insensitivity of the climate-change signal to the tracking method remains consistent (Fig. S6 and accompanying discussion).

Our second main result is that, unlike the tracking algorithm, the definition of what constitutes a convective object has a potentially large impact on the climate-change signal for all object properties, as well as for changes in the total number of objects. The minimum precipitation intensity ($P_{min}$), minimum size ($A_{min}$) and minimum lifetime ($T_{min}$) thresholds for defining a convective object were all found to be relevant. How the climate-change signal responds to varying these thresholds was found to depend on the object property under investigation. For example, the minimum object size had no significant impact on changes in the object's precipitation intensity, but did lead to different climate-change signals for changes in the total number of objects, as well as changes in object properties like the integrated precipitation volume, distance travelled and more. Similarly, the minimum intensity threshold affected the climate-change signal of object intensity, but was not relevant for, e.g., changes in the object volume. Changes in total convective precipitation were also sensitive to how an object is defined. As discussed in the introduction, the definition of what constitutes a convective object shows considerable variance in the literature. An open question in climate-change research is whether the spatial extent of convective storms will increase or decrease with warming (Fowler et al., 2021). Our results suggest that, at least in some regions, the answer may be dependent on how a convective storm is defined. For our case study, the most intense convective objects showed a decrease in spatial extent (-6 %), while less intense objects showed an increase (up to +15 %).





The results for higher quantiles are generally as expected based on those described above for the median. An exception is for the climate-change signal of precipitation intensity as $P_{min}$ increases, which sees a levelling-off at higher thresholds. Otherwise, the main difference is that, in many cases, the uncertainty in the climate-change signal grows, so that the number of statistically significant differences based on different object definitions reduces (Figures S7-S9 and accompanying discussion). Uncertainty due to the higher quantiles would be expected to decrease with a larger sample of convective objects, e.g. from

longer, climate-length, simulations.

To reduce the sensitivity of Lagrangian-based projections to how an object is defined, we suggest performing spectrum-based analysis by first, e.g., binning the data based on object area, intensity and lifetime before computing the desired statistic within each range of interest (Section 5). Using this approach, it was revealed for our study period that under an RCP8.5 scenario convective cells will (in general) increase in number, area and volume, but decrease in mean intensity. The latter will, however,

be of secondary importance and total convective precipitation is projected to increase. For the convective cells with the most extreme mean intensities, however, the total precipitation volume and area will decrease, while the maximum local intensities will increase.

Our results hint that the sensitivity of the climate-change signal to how an object is defined may, for certain (not all) object properties, decline as object size increases (Figs. 4, 6, 8b, S4). Were this the case, then studies focused on larger precipitation

systems (e.g. Nissen and Ulbrich, 2017; Prein et al., 2017) could be expected to lead to higher certainty. This finding, however, cannot automatically be extrapolated to other weather situations or studies at climate timescales and, thus, requires further investigation. It is similarly true that the sensitivities found for our test period would not necessarily be the same sensitivities found in other studies, as our experiment encompasses a specific period, region and climate-change profile. What we have demonstrated, is the principle that in Lagrangian analyses of convective cells, the climate-change signal of different object

properties can be dependent on how an object is defined. This dependency also has consequences for identifying the physical mechanisms underlying future changes in total convective precipitation. The relative importance of specific object properties in interpreting changes in total convective precipitation will not remain constant if these properties' climate-change signals respond differently to changes in how an object is defined. As such, analysing Lagrangian projections by first partitioning the data based on specific object properties (e.g. intensity, area, lifetime) can clarify the underlying mechanisms by which future

precipitation changes.

For researchers studying future changes in convective precipitation using Lagrangian methods, the first message is that, amongst the standard approaches, the choice of tracking algorithm will likely have little impact on the results. The second message is that the minimum thresholds for what constitutes a convective object should be carefully chosen based on what is most appropriate for (1) the study region and (2) the aims of the study. Alongside this, the change signal across a range of

object intensities, areas and lifetimes should be explored (see Fig. 8). While computational resources may limit how low the object thresholds can be set, the lower the object thresholds then the wider the range of responses that can be investigated. To conclude, Lagrangian analysis is an important technique for studying future changes in precipitation. To make best use of this approach, the uncertainties in the climate-change signal associated with how a convective object is defined should be examined wherever possible.



*Code and data availability.* The tracking algorithms were written using NCL version 6.5 (NCL, 2018) and have been deposited (open access) at https://doi.org/10.5281/zenodo.6977074 (Meredith et al., 2022b). The 0.025° simulation data have been archived under an open access license at the DKRZ Long Term Archive with the permanent link <https://www.wdc-climate.de/ui/entry?acronym=DKRZ_LTA_ 1152_ds00302> (Meredith et al., 2022a). The ERA-Interim and MERRA2 reanalyses used as lateral boundary forcing are publicly available via https://www.ecmwf.int/en/forecasts/datasets/reanalysis-datasets/era-interim and https://gmao.gsfc.nasa.gov/reanalysis/MERRA-2/,
respectively (last accessed 01.08.2022).

*Author contributions.* EPM designed the experiment, wrote the algorithms, performed the analysis and wrote the manuscript. UU provided ideas and comments for the analysis and manuscript. HWR commented on the analysis.

*Competing interests.* The authors declare no competing interests.

*Acknowledgements.* This study was funded by the German Ministry of Education and Research (Bundesministerium für Bildung und
Forschung, BMBF) as part of the ClimXtreme project (https://climxtreme.net/). More specifically, the work was performed as part of the ClimXtreme sub-project XPreCCC, grant number 01LP1902H. Computing resources were provided by the German Climate Computing Centre (DKRZ) under project ID bb1152 and by the HPC service of ZEDAT, Freie Universität Berlin (http://dx.doi.org/10.17169/refubium-26754). We thank the CLM-Community (https://www.clm-community.eu/; last accessed 01.08.2022) for maintaining and providing the COSMO-CLM regional climate model. COSMO-CLM is the community model of the German regional climate research community,
jointly further developed by the CLM-Community.



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
