# Peer review of "Cell tracking of convective rainfall: sensitivity of climate-change signal to tracking algorithm and cell definition (Cell-TAO v1.0)"

_Geoscientific Model Development, 2022_

## Referee Comment (RC1)

Review of "Cell tracking of convective rainfall: sensitivity of climate-change signal tracking algorithm and cell definition (Cell-TAO v1.0)" by E. P. Meredith, U. Ulbrich, and H. W. Rust.

Review by Matthew Igel.  Review requested on September 28, 2022. Review completed October 10, 2022.

Summary

A large ensemble of convective-allowing simulations driven by observed meteorology over Germany and a pseudo global warming perturbation are used to test the sensitivity of climate change signals to arbitrary thresholds in Lagrangian cloud-object methods. Choices of arbitrary parametric thresholds are shown to impact the magnitude of the climate change signal while methodological choices are not. Finally, some broad conclusions regarding the climate change signal are drawn.

I think the question of whether arbitrary methodological and parametric choices have an impact of our conclusions regarding climate change is an excellent one.  The simulations used are a reasonable test of this, and the analysis framework is clear and logical. I have concerns about the generality of their simulations to represent *magnitudes* of changes due to climate warming rather than *differences* in changes.  I also wonder how sensitive the conclusions are to the native time resolution of the input data.

Major Comments

1. The lack of sensitivity to the tracking method of the climate change signal is encouraging, but I think more needs to be done to solidify this conclusion.  At 5-minutely data input and 2.8km spacing, the conceptual difference between the overlap and advection methods are very small. It is likely that any horizontally extensive rain object will "overlap" both before and after "advection".  So, in that way, it is unsurprising that these methods yield similar results.  To really test this conclusion, I think some sensitivity tests to input data resolution need to be tested.  For example, 10-minutely and 15-minutely data inputs could be tested.  By 15-minutely data, one might imagine that the redundancy of the methods might be relaxed and the resulting signals might be more different.  Additionally, one could imagine coarsening the input data from 2.8km to 5.6km and 11.2km and then re-running the analysis again.

2. Section 5 feels incongruous with Section 4. As stated in the second paragraph, the results feel too anecdotal (and I would add the microphysics scheme as a probable source of uncertainty) to derive much confidence in the magnitude of the climate change signal. Even focusing on the sign of the changes feels too confident. The experimental setup is an excellent one for showing the sensitivity to arbitrary thresholds, but 2-week simulations cannot be considered representative of climatology.

Minor Comments

L14: I don't follow this sentence.

Introduction: The introduction is clear and appropriately thorough.  It is nicely written.

L100: I had never heard of this technique for creating an ensemble. It seems perfectly reasonable, but why was it chosen for this study? Do you track precipitation only over the overlapping domain? If not, why introduce this complication to data interpretation?

Section 4.2: It might be worth mentioning that even the *sign* of the climate change signal sometimes depends on $A_{min}$.

L214: this is a nice point.

Figure 7: why use different colorbar minima?

Conclusions: some of these will need to be removed (pending the response to Major Concern 2).

---

## Referee Comment (RC2)

**Review of gmd-2022-202**

In this paper the authors test the sensitivity of climate change signals found in simulations of present and future convective rainfall to Lagrangian analysis method and convective cell definition. They test two commonly-used object tracking methods and find that the choice of method makes no significant difference to the resulting climate change signal. The definition of the convective cell tracked by the object-based method, however, is shown to have a significant effect on the changes shown by the analysis. The authors thoroughly analyse these differences in climate change signal and propose that future studies use spectrum-based analyses and consider cell definition carefully.

The paper is admirably clear and well written; the study is thorough and the results highly relevant. The results show the importance of good experimental design and consideration of the limitations of any given analysis technique. The paper is highly polished and scientifically rigorous. I have only a few minor comments that the authors should take into account in a revised manuscript.

**Comments**

1. Figure 1: Some possibly-spurious black dots appear on the edges of sub-plots a) and b). Units are required for the gdpm scale and sea-level pressure.

2. Section 3.1: What spin-up time was used for the simulations?

3. Line 106: The authors should briefly explain their choice of using RCP8.5 as the future pathway in this study.

4. Line 111: Figure 1c includes only specific humidity and temperature, yet the climate change signal mentioned here includes wind and pressure also. The authors should clarify which variables are affected.

5. Figure 2: 'green' in the caption should be 'yellow'.

6. Line 153: While these variables are mentioned in Section 3.2, the fact the authors analyse those variables is only mentioned here. I suggest rephrasing this line.

7. Line 154: The median is calculated yet the change in mean is considered as the climate signal – is this correct, or is the change in median also used as the climate change signal? The authors should clarify.

8. Lines 158-166: When the different thresholds are introduced it would be useful for the authors to include a short description of how they were chosen; for example were they chosen to fit within observed ranges, or around average conditions for the region?

9. Line 166: Absolutely values of which variables or properties?

10. Figures 4 and 5: It seems statistical significance of the signals are shown by using solid or dashed symbols – this should be noted in the Figure captions. This comment goes for all similar plots in the manuscript and supplementary material.

11. Line 226: Should $P_{min}$ here be $P_{avg}$, since $P_{min}$ was held constant when $A_{min}$ was varied?

12. Lines 240-245: It would be useful to include some analysis of the physical meanings of these results – is the reason that the climate change signal is more affected by $T_{min}$ when $A_{min}$ is small that when larger objects tend to have longer lifespans and therefore increasing $A_{min}$ removes objects that would be excluded by increasing $T_{min}$?

13. Figure S6 and discussion: The authors have included a useful analysis of a longer timeseries to see whether their conclusion that the tracking method has no significant effect on climate change signal holds. They comment that the one parameter in which differences are observed is the storm lifespan, the only discritised parameter, and that the discretisation may cause changes to appear large. The authors should note here whether they tested this hypothesis for explaining this difference, by for example looking at the distributions of lifespan values produced by each method instead of only comparing medians.

14. Lines 346-352: The suggestion to use spectrum-based analysis is a good one that follows nicely from this study. However, I presume that the results of spectrum-based analyses also depend somewhat on the definition of the object; for example if the minimum storm size is large then smaller storms will not be included even in the spectrum-based analysis. The authors should comment on this point.

15. Line 368: Related to the previous point; I see the authors' recommendation is to choose thresholds based on the study region and aims of the study. I would also suggest that the thresholds may depend on the model and convective scheme used, and should be chosen to best represent objects in the present climate. One way to do so would be, for example, to compare to radar observations as per Caine et al. (2013) or Raupach et al. (2021).

16. Conclusions: The authors tested tracking algorithms based on advection and object overlap. Do the authors have any comment or hypothesis on the effect of using a pattern-matching algorithm?

**References**

Caine, S., T. P. Lane, P. T. May, C. Jakob, S. T. Siems, M. J. Manton, and J. Pinto, 2013: Statistical assessment of tropical convection-permitting model simulations using a cell-tracking algorithm. *Monthly Weather Review*, **141 (2)**, 557 – 581, doi:10.1175/MWR-D-11-00274.1.

Raupach, T. H., A. Martynov, L. Nisi, A. Hering, Y. Barton, and O. Martius, 2021: Object-based analysis of simulated thunderstorms in switzerland: application and validation of automated thunderstorm tracking with simulation data. *Geosci Model Dev*, **14 (10)**, 6495–6514, doi:10. 5194/gmd-14-6495-2021.

---

## Author Response (AR1)

**Response to Reviewers #1 and #2**

December 23, 2022

We would like to begin by thanking both reviewers for volunteering their time to evaluate our manuscript; this is most appreciated. We are furthermore grateful for the thoughtful and constructive comments. In the following, we respond to each comment individually. Reviewer comments are indented and in *italic font*. Throughout the response, when referencing line numbers, "$L_o$" is used to refer to line numbers in the **o**riginal manuscript, and "$L_r$" used to refer to line numbers in the **r**evised manuscript.

Before the response, I need to report that while acting on one of the comments from Reviewer #1, a small bug was detected in the advection-based tracking algorithm. This affected some specific cases where a track was erroneously ended prematurely after a merger had taken place. On very close inspection, some small differences from the original manuscript can be seen in some figures and Table 1. These are however all minor, statistically insignificant and do not alter the conclusions in any way. The tracking scripts have been updated in the zenodo archive to account for (i) this bug and (ii) making it easier to track data of different temporal resolution.

**1 Response to Reviewer #1**

**Reviewer Summary**

*A large ensemble of convective-allowing simulations driven by observed meteorology over Germany and a pseudo global warming perturbation are used to test the sensitivity of climate change signals to arbitrary thresholds in Lagrangian cloud-object methods. Choices of arbitrary parametric thresholds are shown to impact the magnitude of the climate change signal while methodological choices are not. Finally, some broad conclusions regarding the climate change signal are drawn.*

*I think the question of whether arbitrary methodological and parametric choices have an impact of our conclusions regarding climate change is an excellent one. The simulations used are a reasonable test of this, and the analysis framework is clear and logical. I have concerns about the generality of their simulations to represent magnitudes of changes due to climate warming rather than differences in changes. I also wonder how sensitive the conclusions are to the native time resolution of the input data.*

**Major Comments**

*1. The lack of sensitivity to the tracking method of the climate change signal is encouraging, but I think more needs to be done to solidify this conclusion. At 5-minutely data input and 2.8km spacing, the conceptual difference between the overlap and advection methods are very small. It is likely that any horizontally extensive rain object will "overlap" both before and after "advection". So, in that way, it is unsurprising that these methods yield similar results. To really test this conclusion, I think some sensitivity tests to input data resolution need to be tested. For example, 10-minutely and 15-minutely data inputs could be tested. By 15-minutely data, one might imagine that the redundancy of the methods might be relaxed and the resulting signals might be more different. Additionally, one could imagine coarsening the input data from 2.8km to 5.6km and 11.2km and then re-running the analysis again.*

The suggestion is appreciated. Indeed, analyses have revealed the Reviewer's hypothesis to be correct. I have thus added a new Section 4.6 dealing with the impact of data spatiotemporal resolution. Here, the tracking is re-run using input data temporally aggregated to resolutions of 5-, 15-, 30-, 45- and 60 minutes and, separately, with a spatial resolution aggregated from the original grid to 2x2-, 3x3-, 4x4- and 5x5-cell grids (other settings are held as in the reference setup). The analysis (see new section for more detail) reveals that varying the temporal resolution produces statistically significant differences between the algorithms, which are particularly prominent for the distance travelled and speed metrics. For the other metrics, any statistically significant differences between the tracking methods tend to emerge as temporal resolution decreases, though this doesn't appear to be completely systematic. As for changed spatial resolution, here only isolated cases of statistically significant differences were found between the algorithms. Maybe these would disappear with a larger sample size (coarsening the grid reduces the number of convective objects).

The former result raised the suspicion that, with different temporal resolutions, the differences between the tracking methods described above may reduce (or disappear) with larger objects and increase with smaller objects. As a result, I repeated the temporal-resolution analysis with the largest ($A_{min} = 64$ grid boxes) and smallest ($A_{min} = 2$ grid boxes) minimum-area thresholds. The results confirmed the suspicion, and this is now discussed in the manuscript. The revised manuscript contains a figure for $A_{min} = 8$ grid boxes, while the same figures using the minimum and maximum $A_{min}$ thresholds are in the supplementary information. The results for the changed spatial resolution are also shown in the supplementary information.

How do these findings affect the conclusion that the tracking method has no impact on the climate-change signal? I would say that the conclusion needs to be revised with the following caveat:

- If the minimum-area threshold for an object is small, then the temporal resolution of the input data must be higher. Otherwise, differences in the climate-change signal can emerge between tracking methods.

Or, perhaps more broadly: threshold choices which serve to increase inhomogeneity of the precipitation field – and, hence, the fraction of objects with a low spatial extent – require input data with a higher temporal resolution. I've tried to comminicate this in the new Section 4.6.

> *2. Section 5 feels incongruous with Section 4. As stated in the second paragraph, the results feel too anecdotal (and I would add the microphysics scheme as a probable source of uncertainty) to derive much confidence in the magnitude of the climate change signal. Even focusing on the sign of the changes feels too confident. The experimental setup is an excellent one for showing the sensitivity to arbitrary thresholds, but 2-week simulations cannot be considered representative of climatology.*

I accept this point. The original motivation behind Section 5 ("Results II: Future projections") was to consider how Lagrangian analysis could be performed/presented in such a way that the results might be less sensitive to certain arbitrary thresholds. For this reason, I would like to retain Section 5 in some form. In the original text I explicitly stated that in Section 5 the magnitude of the climate-change signal was to be examined (e.g. $L_o$75, 262). With hindsight, and in light of the Reviewer's comment, this was a misguided aim. In the revised version, I have retained Section 5 and Figure 8, while making the following changes:

1. Changed the title of the section from "Results II: ..." to "Analysis of future projections". Hopefully, this will point to the aim not being to determine the magnitude of future projections (i.e. declaring a "result"), but rather to suggest a useful way in which Lagrangian projections can be analysed to reduce the sensitivities found in the preceding section

2. Removed references to "magnitude" from the Introduction and the start of Section 5.

3. In the revised section, percentage changes are no longer reported. Instead, aspects of the warming signal are qualitatively reported – i.e. nonlinear, etc. – in the context of demonstrating insights from such an approach as compared to those gained from the approach in Section 4.

4. Re-emphasize that the results are for a case study and are not climatologically representative. The results should be considered as illustrative and only representative for the specific conditions present during our cast study ($L_r$297-299).

I feel that the type of examples in Section 5 are relevant for illustrating the utility of Lagrangian climate-change analysis. Especially in light of Section 4, I don't want to give the impression that projections based on cell tracking are useless.

**Minor Comments**

> *L14: I don't follow this sentence.*

Here I was trying to say that testing the sensitivity of the results to the tracking scheme (i.e. tracking method, thresholds, etc.) is a valuable addition to the overall analysis. As the new analysis of the impact of the input data's spatiotemporal resolution requires further space in the abstract, I have decided to just delete the sentence in question as it's anyway implied by the preceding sentences and is discussed later in the manuscript.

> *Introduction: The introduction is clear and appropriately thorough. It is nicely written.*

Thanks!

*L100: I had never heard of this technique for creating an ensemble. It seems perfectly reasonable, but why was it chosen for this study? Do you track precipitation only over the overlapping domain? If not, why introduce this complication to data interpretation?*

The technique wasn't chosen for any particular reason other than that it's quick, simple and the authors had previous experience applying it (e.g. Noyelle et al., 2018). Please note, however, that the domain shifting is only performed on the parent 0.11° simulation domain. The tracking, on the other hand, is only performed on the 0.025° domain, which is fixed in space – i.e. it does not get shifted with the 0.11° domain. I had tried to highlight this ($L_o$113) by referring to a "geographically fixed" domain. In the revised version, I have attempted to emphasize this point by (1) giving the original reference to "geographically fixed" its own sentence beginning with "Note that" ($L_r$116), and (2) re-emphasizing that the analysis is over the (fixed) COSMO-DE domain at the start of the Analysis description ($L_r$151).

*Section 4.2: It might be worth mentioning that even the sign of the climate change signal sometimes depends on $A_{min}$.*

Done! ($L_r$214)

*L214: this is a nice point.*

Thanks!

*Figure 7: why use different colorbar minima?*

The reason for this was that, while the four different panels had similar maxima, their minima were quite different. Thus, using a single colour bar which fully incorporated the ranges of all four panels led – in some cases – to the differences in changes being hard to discern, i.e. the gradients across the matrix were not so apparent; this was a particular issue when jointly varying $A_{min}$ and $T_{min}$ (panels a and c). I've illustrated this issue in Fig. R1, where the colour bar has a uniform range of 70–120 %, which encompasses all four minima. In my opinion, this is of sub-optimal appearance (for the reasons stated before).

Having said that, adopting four different colour bars may have been overkill. I have thus decided to seek a middle ground, in which a single colour range is used for jointly varying object size and lifetime (panels a and c) and another single colour range for jointly varying object intensity and lifetime (panels b and d). This is illustrated in Fig. R2 and now features in the revised manuscript.

If the Reviewer disagrees with this, I would be prepared to use a uniform colour bar as in Fig. R1.

*Conclusions: some of these will need to be removed (pending the response to Major Concern 2).*

Done.

[Figure]

Figure R1: As in Fig. 7 of the original manuscript, except using a uniform colour bar of 70 − 120 % for all panels.

[Figure]

Figure R2: As in Fig. 7 of the original manuscript, except using a uniform colour bar of 84 – 120 % for panels a and c (joint variation of $T_{min}$ and $A_{min}$), and of 70 – 120 % for b and d (joint variation of $T_{min}$ and $P_{min}$.)

**2 Response to Reviewer #2**

*In this paper the authors test the sensitivity of climate change signals found in simulations of present and future convective rainfall to Lagrangian analysis method and convective cell definition. They test two commonly-used object tracking methods and find that the choice of method makes no significant difference to the resulting climate change signal. The definition of the convective cell tracked by the object-based method, however, is shown to have a significant effect on the changes shown by the analysis. The authors thoroughly analyse these differences in climate change signal and propose that future studies use spectrum-based analyses and consider cell definition carefully.*

*The paper is admirably clear and well written; the study is thorough and the results highly relevant. The results show the importance of good experimental design and consideration of the limitations of any given analysis technique. The paper is highly polished and scientifically rigorous. I have only a few minor comments that the authors should take into account in a revised manuscript.*

**Comments**

*1. Figure 1: Some possibly-spurious black dots appear on the edges of sub-plots a) and b). Units are required for the gdpm scale and sea-level pressure.*

Thanks, the black dots were an artefact of overly-thick national boundaries used in the map; this has now been rectified. With regards to the units, I hope I've understood correctly: I just added 'gpdm' and 'hPa' to the figure caption.

*2. Section 3.1: What spin-up time was used for the simulations?*

The 0.11° and 0.025° simulations were initialized at 00:00 UTC on 26.05.2016 and 27.05.2017, respectively. The analysis period begins on 27.05.2016 at 04:00 UTC in the 0.025° model, giving 4 h spinup in the CPM. I've added this to Section 3.1 ($L_r$115).

"Spin-up" here is supposed to refer to the spinning-up of small-scale (convective) precipitation features in the CPM. For example, in a downscaling from 15 km to 3 km (we have 12 km to 2.8 km) over the central United States, Wong and Skamarock (2016, Fig. 2b) found a value of about 6–12 h for the spin-up of small-scale convective precipitation features. In our convection-permitting simulations, the analysis period begins 4 h after the CPM is initialised from its 0.11° parent model. Fig. R3 illustrates the rapid adjustment of small-scale precipitation features in the CPM after initialisation. Here we see the temporal evolution of the spatial standard deviation of (i) column-integrated cloud graupel, (ii) column-integrated cloud ice, and (iii) column-integrated cloud water. Note that these variables (on model levels) were all included in the initial conditions provided by the 0.11° model. A high standard deviation would represent high small-scale variability of these variables. In the case of (a) cloud graupel and (c) cloud water, there is a rapid increase in small-scale variability after initialization, with the period of rapid growth completed before the start of the analysis period. In the case of (b) cloud ice, there is a rapid decrease in small-scale variability which is driven by the removal of cloud ice present in the initial conditions provided by the 0.11° model (not shown); the period of rapid adjustment ends before the analysis period. For this reason, I believe that the spin-up prior to the analysis period is sufficient.[1]

*3. Line 106: The authors should briefly explain their choice of using RCP8.5 as the future pathway in this study.*
* * *
[1]Note that in a recent paper in NHESS which I co-authored, I received a similar reviewer comment and my response in this paragraph is very similar to my response there (just in case it looks like I've copied from another source!); please see the online discussion of Caldas-Alvarez et al. (2022)

[Figure]

Figure R3: Temporal evolution from initialisation for the spatial standard deviation of (i) column-integrated cloud graupel, (ii) column-integrated cloud ice, and (iii) column-integrated cloud water. The curves represent the 0.025° convection-permitting model, which was initialized from its 0.11° parent model at 0000 UTC on 27[th] May 2016. The analysis period – marked with a vertical black line – starts at 0400 UTC on 27[th] May 2016, four hours later. The spatial standard deviation is computed over the analysis region shown in Figures 1 and 3 of the manuscript.

The scenario was chosen simply because it is a high-end scenario and thus likely to have a clear climate-change signal to serve as a basis for our experiment. I've added this to the text (L$_r$106-107).

> *4. Line 111: Figure 1c includes only specific humidity and temperature, yet the climate change signal mentioned here includes wind and pressure also. The authors should clarify which variables are affected.*

It is correct that pressure and winds are also varied. I've added the change in winds (on pressure levels), as well as pressure, winds, temperature and humidity (all on height levels) to the supplementary information. This supplementary figure is now referenced to in the caption of Figure 1. I prefer to place the additional figures in the supplementary information as, in my opinion, they are of secondary importance to the temperature and humidity (as shown in Fig. 1c) and I also don't want to over-clutter Fig. 1.

> *5. Figure 2: 'green' in the caption should be 'yellow'.*

Done. Thanks for spotting this error.

> *6. Line 153: While these variables are mentioned in Section 3.2, the fact the authors analyse those variables is only mentioned here. I suggest rephrasing this line.*

Done. (L$_r$154).

> *7. Line 154: The median is calculated yet the change in mean is considered as the climate signal – is this correct, or is the change in median also used as the climate change signal? The authors should clarify.*

For each member we compute the 0.5 quantile of each object characteristic. The ensemble mean of these values is then computed, before being compared with the same ensemble mean of the other climate. In the supplementary information, we do the same for the 0.9 quantile. I've tried to reword the passage so that it is less confusing (L$_r$155-157).

> *8. Lines 158-166: When the different thresholds are introduced it would be useful for the authors to include a short description of how they were chosen; for example were they chosen to fit within observed ranges, or around average conditions for the region?*

I think the original way I wrote this paragraph misrepresented the chronological order. In reality, the reference settings were defined before the experiment and the thresholds varied afterwards. I've now changed the paragraph to first define the reference settings and the reasons for choosing them. Then, the variation of the thresholds is explained as being symmetric about the reference settings, or increased from $T_{min}$ (due to the reference there already being low).

> *9. Line 166: Absolutely values of which variables or properties?*

"Absolute values" was a poor choice of words. I wanted to say that raw values from the present and PGW ensembles which underlie the climate-change signal can also be found in Table 1. To resolve this, I've decided to simply remove the second half of the sentence – ", where the absolute values can also be found" – as it's anyway somewhat superfluous.

> *10. Figures 4 and 5: It seems statistical significance of the signals are shown by using solid or dashed symbols – this should be noted in the Figure captions. This comment goes for all similar plots in the manuscript and supplementary material.*

I've modified the captions. Hopefully it's clear now.

*11. Line 226: Should Pmin here be Pavg, since Pmin was held constant when Amin was varied?*

Yes, thanks for spotting!

*12. Lines 240-245: It would be useful to include some analysis of the physical meanings of these results – is the reason that the climate change signal is more affected by Tmin when Amin is small that when larger objects tend to have longer lifespans and therefore increasing Amin removes objects that would be excluded by increasing Tmin?*

I think it's because raising $A_{min}$ increases the proportion of larger objects in the sample: larger objects also tend to live longer. As such, in a sample of longer-lived objects, the climate-change signal is less affected by (relatively) small changes in $T_{min}$. For (an exaggerated) example, if all of the objects in the sample live for at least 90 minutes, then whether $T_{min}$ is 15-, 30-, 45-, or 60 minutes is of no consequence to the climate-change signal. I've added a few sentences to this effect (L$_r$250-254).

*13. Figure S6 and discussion: The authors have included a useful analysis of a longer timeseries to see whether their conclusion that the tracking method has no significant effect on climate change signal holds. They comment that the one parameter in which differences are observed is the storm lifespan, the only discritised parameter, and that the discretisation may cause changes to appear large. The authors should note here whether they tested this hypothesis for explaining this difference, by for example looking at the distributions of lifespan values produced by each method instead of only comparing medians.*

I've added a new Q-Q plot to the discussion which illustrates how the discretized nature of the lifetime metric can lead to large differences in the climate-change signal between similar quantiles. The Q-Q plot is repeated for all other object properties, revealing the unique nature of the lifetime metric in this respect.

*14. Lines 346-352: The suggestion to use spectrum-based analysis is a good one that follows nicely from this study. However, I presume that the results of spectrum-based analyses also depend somewhat on the definition of the object; for example if the minimum storm size is large then smaller storms will not be included even in the spectrum-based analysis. The authors should comment on this point.*

I've added a few sentences about this to the paragraph (L$_r$373-375). Correspondingly, a sentence covering similar ideas has been removed from the final paragraph to avoid duplication.

*15. Line 368: Related to the previous point; I see the authors' recommendation is to choose thresholds based on the study region and aims of the study. I would also suggest that the thresholds may depend on the model and convective scheme used, and should be chosen to best represent objects in the present climate. One way to do so would be, for example, to compare to radar observations as per Caine et al. (2013) or Raupach et al. (2021)*

Thanks for the suggestion. I've added a sentence on this to the final paragraph, but have refrained from expanding on the point in detail as I would like to keep the final paragraph as concise as possible.

*16. Conclusions: The authors tested tracking algorithms based on advection and object overlap. Do the authors have any comment or hypothesis on the effect of using a pattern-matching algorithm?*

I think that the conclusion that the tracking algorithm doesn't impact the climate-change signal would also apply to the pattern-matching approach and I've added a sentence to this effect (L$_r$348-350); the reasoning behind this conclusion stems from the new analysis in Section 4.6. If this is true, then it would

follow that the pattern-matching approach would also show the same sensitivities to the threshold choices as the other algorithms.

**References**

S. Caine, T. P. Lane, P. T. May, C. Jakob, S. T. Siems, M. J. Manton, and J. Pinto. Statistical Assessment of Tropical Convection-Permitting Model Simulations Using a Cell-Tracking Algorithm. *Monthly Weather Review*, 141(2):557–581, 2013. doi: 10.1175/MWR-D-11-00274.1.

A. Caldas-Alvarez, M. Augenstein, G. Ayzel, K. Barfus, R. Cherian, L. Dillenardt, F. Fauer, H. Feldmann, M. Heistermann, A. Karwat, F. Kaspar, H. Kreibich, E. E. Lucio-Eceiza, E. P. Meredith, S. Mohr, D. Niermann, S. Pfahl, F. Ruff, H. W. Rust, L. Schoppa, T. Schwitalla, S. Steidl, A. H. Thieken, J. S. Tradowsky, V. Wulfmeyer, and J. Quaas. Meteorological, impact and climate perspectives of the 29 June 2017 heavy precipitation event in the Berlin metropolitan area. *Natural Hazards and Earth System Sciences*, 22(11):3701–3724, 2022. doi: 10.5194/nhess-22-3701-2022. URL `https://nhess.copernicus.org/articles/22/3701/2022/`.

R. Noyelle, U. Ulbrich, N. Becker, and E. P. Meredith. Assessing the impact of SSTs on a simulated medicane using ensemble simulations. *Natural Hazards and Earth System Sciences Discussions*, 2018:1–23, 2018. doi: 10.5194/nhess-2018-230. URL `https://www.nat-hazards-earth-syst-sci-discuss.net/nhess-2018-230/`.

T. H. Raupach, A. Martynov, L. Nisi, A. Hering, Y. Barton, and O. Martius. Object-based analysis of simulated thunderstorms in Switzerland: application and validation of automated thunderstorm tracking with simulation data. *Geoscientific Model Development*, 14(10):6495–6514, 2021. doi: 10.5194/gmd-14-6495-2021.

M. Wong and W. C. Skamarock. Spectral characteristics of convective-scale precipitation observations and forecasts. *Monthly Weather Review*, 144(11):4183 – 4196, 2016. doi: 10.1175/MWR-D-16-0183.1. URL `https://journals.ametsoc.org/view/journals/mwre/144/11/mwr-d-16-0183.1.xml`.